# An Underrecognized Histologic Clue to the Diagnosis of Mucous Membrane Pemphigoid: A Case Report and Review of Diagnostic Guidelines

**Jason R. McFadden** [1,*,†], **Advaita S. Chaudhari** [1,†], **Debra Birenbaum** [2,3], **Lynette Margesson** [2,3], **Jorge Gonzalez** [2] and **Aravindhan Sriharan** [2,4]

1 Department of Biological Sciences, Dartmouth College, Hanover, NH 03755, USA
2 Department of Pathology & Laboratory Medicine, Dartmouth-Hitchcock Medical Center, Lebanon, NH 03766, USA
3 Department of Obstetrics & Gynecology, Dartmouth-Hitchcock Medical Center, Lebanon, NH 03766, USA
4 Department of Pathology & Laboratory Medicine, Geisel School of Medicine at Dartmouth, Hanover, NH 03755, USA
* Correspondence: jason.mcfadden@nih.gov; Tel.: +1-240-778-5146
† These authors contributed equally to this work.

**Abstract:** Mucous membrane pemphigoid (MMP), also known as cicatricial pemphigoid (CP), is a heterogeneous group of subepidermal blistering diseases that affect the mucous membranes, most frequently in the eye and oral cavity. MMP is often unrecognized or misdiagnosed in its early stages due to its rarity and nonspecific presentation. We present the case of a 69-year-old female in which MMP of the vulva was not initially suspected. The first biopsy, from lesional tissue for routine histology, revealed fibrosis, late-stage granulation tissue, and nonspecific findings. A second biopsy, from perilesional tissue for direct immunofluorescence (DIF), revealed DIF findings typical of MMP. Scrutiny of both the first and second biopsies revealed a subtle but telling histologic feature: subepithelial clefts along adnexae in the context of a scarring process with neutrophils and eosinophils, which can be an important clue to MMP. This histologic clue has been previously described; reinforcing its importance may prove useful for future cases, especially those for which DIF is not feasible. Our case demonstrates the protean presentations of MMP, the need for persistence in sampling unusual cases, and the relevance of inconspicuous histologic features. The report highlights this underrecognized yet potentially decisive histologic clue to MMP, reviews current biopsy guidelines when MMP is suspected, and delineates the clinical and morphological features of vulvar MMP.

**Keywords:** mucous membrane pemphigoid; subepithelial clefts along adnexae; gynecological pathology; cicatricial pemphigoid

## 1. Introduction

Mucous membrane pemphigoid (MMP), also known as cicatricial pemphigoid (CP), is a heterogeneous group of subepidermal blistering diseases that affect the mucous membranes and skin [1]. Mucous membranes of the eye and oral cavity are most frequently affected (in 85% and 65% of cases, respectively), followed by mucous membranes of the nose (20–40%), anogenital area (20%), larynx (5–15%), and esophagus (5–15%) [1]. MMP was originally named "benign mucous membrane pemphigoid" to distinguish it from bullous pemphigoid. However, this name was deemed misleading since MMP can have serious long-term sequelae if left untreated, including blindness, conjunctivitis, subconjunctival fibrosis, supraglottic stenosis, and airway obstruction [2]. MMP often goes unrecognized in its early inflammatory stages, both because its initial presentation can be nonspecific and because MMP is a rare disease [1].

## 1.1. Immunofluorescence Studies

Typical direct immunofluorescence (DIF) features of MMP include linear deposition of IgG and C3 along the dermoepidermal junction. Some cases feature linear deposition of IgA as well. The target antigens can include BP230 (anti-BP230-type MMP) and BP180 (type XVII collagen; anti-BP180-type MMP), two hemidesmosomal proteins located in the epithelial basement membrane zone (BMZ) (Figure 1). Other targeted BMZ antigens can include laminin-6, type VII collagen, p200 (antilaminin-γ-1 pemphigoid), integrin α6-β4 (oral and ocular MMP), and laminin-332 (antilaminin-332 MMP, previously known as "antilaminin-5 MMP" or "epiligrin MMP") [1,3,4]. The anti-BMZ autoantibodies produced by MMP patients typically bind to BP180′s C-terminal domain, whereas those produced by BP patients typically bind to BP180′s NC16A domain. In the context of MMP, indirect immunofluorescence (IIF) is used to detect circulating anti-BMZ autoantibodies in a patient's serum. For MMP patients, circulating anti-BMZ antibodies are less consistently detected, and the titer of circulating antibodies is considerably lower than that of BP patients.

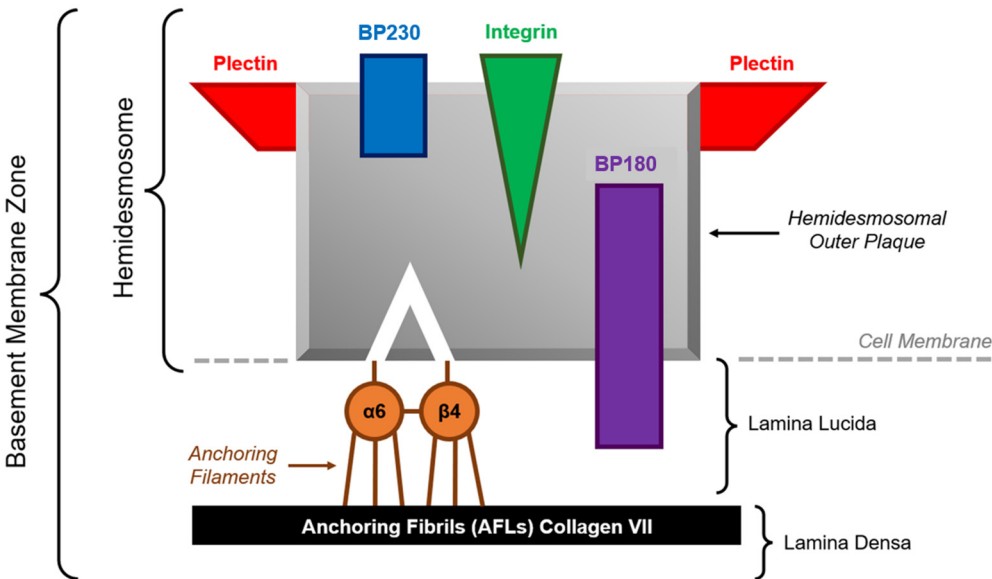

**Figure 1.** Serology for anti-BP230 and anti-BP180 autoantibodies is characteristic of the various types of BP, including different types of MMP. Serology can be useful to rule out other differential diagnoses. Figure adapted from Xu et al [1].

## 1.2. Diagnostic Guidelines

Patients suspected of having MMP typically undergo 2 biopsies: one specimen from the interface of lesional tissue with uninvolved skin, and another specimen from perilesional tissue, for direct immunofluorescence (DIF) studies. DIF specimens solely from the center of the lesion can lead to false-negative interpretations (i.e., missed diagnoses) of MMP as a result of "loss of immunoreactants in longstanding lesions" and lack of immunopathologic specificity" [1].

Regarding the ideal biopsy site and technique, international consensus [5] recommends:

- For single mucous membrane involvement without skin involvement: Perilesional biopsy (i.e., obtain specimen from a location adjacent to the inflamed site).
- For multiple mucous membrane involvements without skin involvement: Obtain specimen from "tissue adjacent to an inflamed nonocular mucosal site".
- For mucous membrane involvement and skin involvement: Perilesional biopsy (i.e., obtain specimen from a location adjacent to the inflamed site).

Such samples should be taken in addition to the lesional biopsy for hematoxylin and eosin (H&E) staining, when possible. H&E biopsies may not be reasonable in all cases (e.g.,

cases of only ocular involvement, in which the act of biopsy can promote scarring that leads to blindness). If biopsy is performed in these rare circumstances, DIF should be prioritized.

Though advanced histologic, immunofluorescence, immunochemical, and immuno-genetic techniques (e.g., detection of the HLA-DQB1*0301 allele in ocular MMP) have been proposed to distinguish between the various subtypes of MMP, diagnosis should be based on clinical presentation and histology [6]. The subtypes of MMP are then delineated by anatomic distribution.

## 2. Case Presentation

A 69-year-old female presented with desquamation around the edges of the labia majora and inner thighs, as well as scarring of the vulva, with slight erosions by the clitoris and significant erosions on the upper perineum and perianal area (Figure 2). Neither intramuscular triamcinolone nor Clobetasol 0.05% ointment relieved this vulvar discomfort. The patient underwent an initial lesional biopsy from the right perianal area, which revealed fibrosis and late-stage granulation tissue. The differential diagnosis at the time of biopsy included lichen planus (LP), lichen sclerosus (LS), differentiated vulvar intraepithelial neoplasia (d-VIN), and MMP. Features of LP, LS, and d-VIN were not evident in the biopsy, ruling out those differential diagnoses. Similarly, evidence of herpes simplex virus (HSV) infection was not identified in the biopsy.

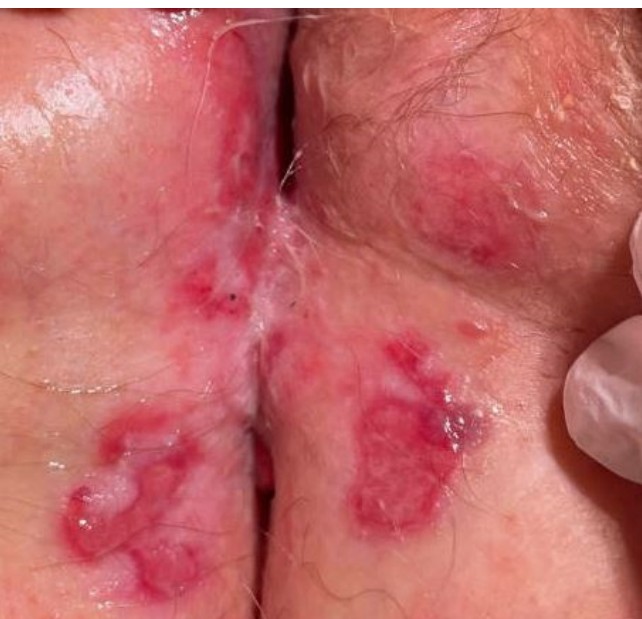

**Figure 2.** Physical examination revealed sharply delineated erosions and scarring in the perineum, typical of mucous membrane pemphigoid. The patient's pruritus and bullae are characteristic of MMP, which is often preceded by a prodromal phase of eczematous rash. Bullae can sometimes rupture to form erosions, as in this case. A potential pitfall is that when the histology consists of fibrosis and late-stage granulation tissue, the findings may be misleading.

One month later, after another 90 mg intramuscular triamcinolone injection, another biopsy was taken from the interface of normal and lesional skin. This perilesional biopsy revealed a subepidermal cell-poor vesicular dermatitis, with neutrophils, eosinophils, and—tellingly—subepithelial clefts that extended along adnexal epithelia (Figure 3). Direct immunofluorescence (DIF) studies revealed linear IgG and C3 deposition along the dermoepidermal junction (Figure 4). Taken together, these findings were consistent with vulvar MMP. The patient's condition has ameliorated since starting treatment.

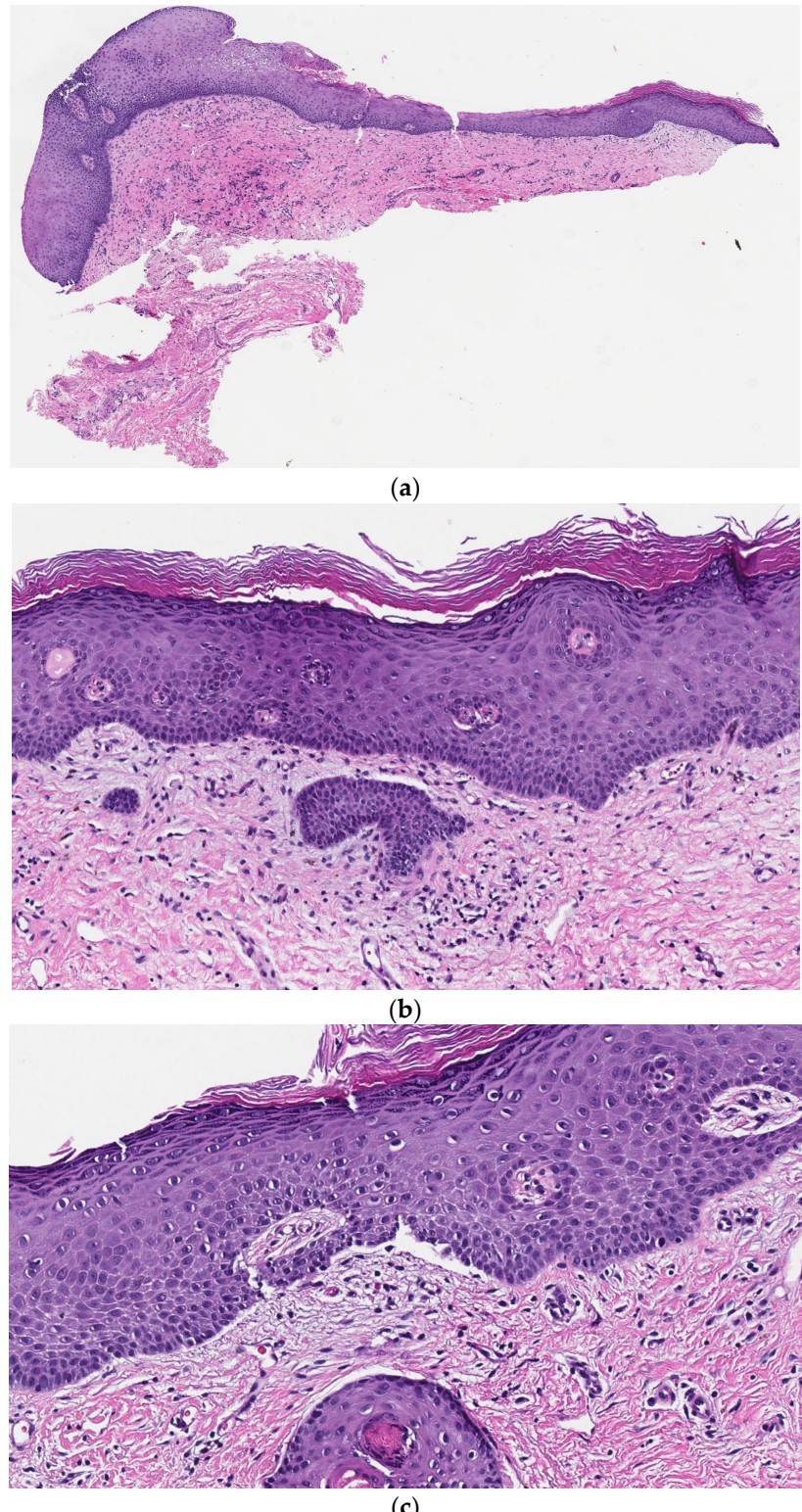

**Figure 3.** Representative histologic images from the first (lesional) biopsy. (**a**) (Hematoxylin and Eosin, 20×). At low power, there are multiple features of a scar. (**b**) (Hematoxylin and Eosin, 100×) and (**c**) (Hematoxylin and Eosin, 1200×). At higher power, there are multiple disparate areas with subtle subepithelial clefting, which involve the adnexae, and are joined by eosinophils and some neutrophils (not shown). Fibrosis and late-stage granulation tissue are also present.

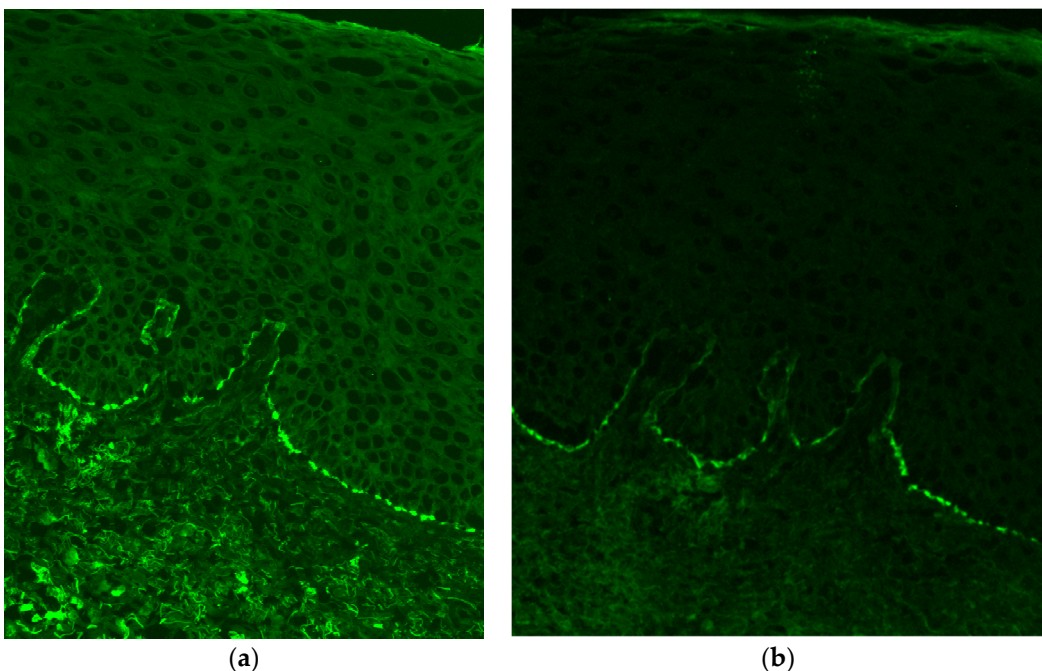

(**a**) (**b**)

**Figure 4.** Linear deposition of (**a**) IgG (4-A, 200×) and (**b**) C3 (4-B, 200×) along the dermoepidermal junction. DIF IgG and DIF C3 stains were used.

## 3. Discussion

Though the various subtypes of bullous disease can share similar histological features, such as prominent fibrosis, the findings from the presented case are consistent with MMP. The patient appeared to have erosive vulvar dermatitis that extended to the perineum, which is suspicious for MMP. However, differential diagnoses such as LP and LS can have similar clinical appearances as MMP, especially when eroded. LP and LS are best ruled out by a biopsy from the interface of normal and eroded skin. While histology can be useful in ruling out other differential diagnoses, the findings are often nonspecific.

After a diagnosis of MMP was established, the first perianal biopsy was reexamined. Therein, subepithelial clefts that extend along the adnexal epithelia were found. While subtle, it has been written that this histologic finding, "particularly in a subepidermal blister that houses neutrophils and sometimes eosinophils, is virtually diagnostic for cicatricial pemphigoid" [7]. Our case illustrates that such a finding can be obscured when there is extensive granulation tissue. In such cases, we again emphasize the importance of clinical suspicion: if either the pathologist or the clinician is vigilant about the possibility of MMP, detection of subepithelial clefts along the adnexae can guide the pathologist to the correct diagnosis. The bullae and inflammatory infiltrates of MMP frequently extend along adjacent epithelial structures; those of BP and other vesiculobullous mimickers generally do not. Importantly, while subepidermal clefts can be seen in a broad range of diseases, the finding of subepidermal clefts specifically associated with adnexae has been described as a typical finding of MMP [7]. Unlike subepidermal clefting generally, subepithelial clefts associated with adnexae can be fairly specific for the diagnosis, in this setting.

In the absence of adjunctive clinical information, the histologic differential diagnosis can also include a scar. However, the trifecta of neutrophils, eosinophils and subepithelial clefts that extend along adnexae are not commonly seen in many scars. In this case, the MMP diagnosis was secured by clinical findings and DIF studies, and was confirmed retrospectively by histopathology. In future cases, especially those wherein DIF is not available to the pathologist, subepithelial clefting associated with adnexae could be a useful clue in this context.

It is imperative to correctly distinguish MMP from its differentials, as untreated lesions can form adhesions that require surgical division. The consequences of both missed MMP

diagnoses (i.e., withholding of immunosuppressive therapy) and incorrect MMP diagnoses (i.e., administration of inappropriate topical therapy or unnecessary adjuvant regimens) are far-reaching. The scar-prone nature of vulvar MMP can lead to labial fusion and introital shrinkage, along with further sexual and urinary problems [8]. Furthermore, the aggressive nature of MMP limits treatment options in advanced stages, as the condition is chronic and progressive. Diagnosis and intervention become significantly more difficult in the later stages of MMP, leading to increased costs and decreased quality of life. The goal of MMP treatment is to reduce inflammation and prevent progression of scarring processes [9]. Early intervention with oral and topical corticosteroids can stop blister formation, promote healing, and prevent scarring [8].

Finally, while the histopathological picture of subepithelial clefting along adnexae, especially in a scarring process with neutrophils and eosinophils, can be useful, DIF for anti-BMZ autoantibodies remains the gold standard for the diagnosis of MMP [9,10]. Furthermore, a single negative DIF result does not fully exclude MMP; in such instances, a repeat biopsy should be considered [10]. It has been shown that multiple and repeated biopsies increase the sensitivity of DIF tests for MMP compared to single biopsies [10]. Histopathology and DIF should be considered together when establishing the diagnosis [9]. As mentioned earlier, histopathology often reveals nonspecific ulcerative inflammation with scarring and granulation tissue. If this is the case, then histopathology alone cannot differentiate MMP from its differential diagnoses, nor can it distinguish between the various subtypes of MMP [9]. Diagnosis of MMP based solely on any single clinical or histologic feature is inconsistent with the international consensus statements [5,10].

## 4. Conclusions

Our case of vulvar MMP in a 69-year-old woman illustrates the protean presentations of MMP. Dermatopathologists and gynecological pathologists should be aware that MMP can have a subtle presentation, mimicking nonspecific reactive changes. But the finding of subepithelial clefts that extend along adnexae, in the context of a scarring process on vulvar skin with neutrophils and eosinophils, can be a clue to guide the pathologist toward the correct diagnosis of cicatricial pemphigoid. Importantly, this histopathologic picture remains secondary to DIF testing for anti-BMZ autoantibodies, the gold standard for the diagnosis of MMP.

**Author Contributions:** Conceptualization, A.S.; methodology, A.S.; formal analysis, D.B., L.M. and J.G.; resources, D.B., L.M. and J.G.; writing—original draft preparation, J.R.M., A.S.C. and A.S.; writing—review and editing, J.R.M., A.S.C. and A.S.; visualization, A.S.; supervision, A.S.; project administration, A.S.; funding acquisition, A.S. All authors have read and agreed to the published version of the manuscript.

**Funding:** This research received no external funding.

**Institutional Review Board Statement:** This study was reviewed and approved by the institutional review board at Dartmouth Health System (IRB Number: 00031828).

**Informed Consent Statement:** Written informed consent has been obtained from the patient to publish this paper.

**Data Availability Statement:** The data in this study were generated through the Clinical Genomics and Advanced Technology (C.G.A.T.) laboratory in the Department of Pathology and Laboratory Medicine of the Geisel School of Medicine at Dartmouth and Dartmouth-Hitchcock Medical Center.

**Acknowledgments:** The authors would like to thank the members of the Pathology Shared Resource Laboratory, a section of the laboratory for Clinical Genomics and Advanced Technology.

**Conflicts of Interest:** The authors declare no conflict of interest.

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
