# Peer review of "An Underrecognized Histologic Clue to the Diagnosis of Mucous Membrane Pemphigoid: A Case Report and Review of Diagnostic Guidelines"

_dermatopathology, doi:10.3390/dermatopathology10010009_

Round 1

Reviewer 1 Report

The paper reports on a case of cicatricial pemphigoid of the vulva and proposes an interesting clue. The problems of histologic diagnosis of this subtype are reviewed. The paper is well written and shows good photographs.

Some minor changes may improve the paper.

Page 1, line 20: add MMP…of the vulva.

Page 1, line 24: say epithelia …in the context of a scarring process with neutrophils and eosinophils can be an important clue to MMP..

Page 2, line 53: explain other types of MMP such as p200- (laminin gamma-) type since this has prognostic and therapeutic implications.

Page 3, line 85: add histology …and Immunofluorescence

Page 3, line 93: explain the abbreviation “d-VIN”

Page 3, line 104: delete the comment “This section may be divided by subheadings… “

Author Response

Response to Reviewers’ Comments

Thank you for giving us the opportunity to submit a revised draft of the manuscript “An Underrecognized Histologic Clue to the Diagnosis of Cicatricial Pemphigoid: A Case Report and Review of Diagnostic Guidelines” for publication. We are grateful for the reviews provided by each external editor and appreciative of the time that each reviewer dedicated to providing insightful feedback on our manuscript.  The comments are encouraging and the reviewers appear to share our judgment that this case and its findings are clinically important. We have incorporated the suggestions made by the reviewers. Please see below, in red, our detailed response to each comment. All page numbers refer to the manuscript file with tracked changes. 

Reviewers' Comments to the Authors:

Responses to Reviewer 1

The paper reports on a case of cicatricial pemphigoid of the vulva and proposes an interesting clue. The problems of histologic diagnosis of this subtype are reviewed. The paper is well written and shows good photographs.

Some minor changes may improve the paper.

  1. Page 1, line 20: add MMP…of the vulva.
  2. Page 1, line 24: say epithelia …in the context of a scarring process with neutrophils and eosinophils can be an important clue to MMP..
  3. Page 2, line 53: explain other types of MMP such as p200- (laminin gamma-) type since this has prognostic and therapeutic implications.
  4. Page 3, line 85: add histology …and Immunofluorescence
  5. Page 3, line 93: explain the abbreviation “d-VIN”
  6. Page 3, line 104: delete the comment “This section may be divided by subheadings… “

Response to Comments A-F: We agree and have incorporated all of your suggestions.

 In particular, regarding suggestion C, we have added 5 additional BMZ antigens that can be targeted in other MMP subtypes: laminin-6, type VII collagen, integrin α6/β4 (oral and ocular MMP), p200 (anti-laminin gamma-1 pemphigoid), and laminin 332 (anti-laminin 332 MMP). Please find our changes on page 2, lines 49-52. Thank you so much!

Reviewer 2 Report

In this paper, the authors report the case of a woman with mucous membrane pemphigoid (MMP) that was initially difficult to diagnose and propose some histologic clues for diagnosis, although the diagnosis was finally confirmed by DIF.

I have the following comments about this manuscript:

 - Mucous membrane pemphigoid (MMP) is NOT a type or variant of cicatricial pemphigoid (CP). In fact, they are synonymous, but some years ago it was decided to use the term MMP instead of CP, as not all cases/locations present with scarring.

- Patients with MMP can target other antigens besides BP180 (type XVII collagen), and BP230 antigens (the terms BPAG1 and BPAG2 are not currently in use) including laminin 332, collagen VII, or integrin alfa6beta4.

- Many patients with MMP are misdiagnosed or not diagnosed initially due to the lack of knowledge about the disease because it is rare. It is not unusual to see patients that have been seen by many physicians (general practitioners, dentists, ENT, gynecologists, etc.) before a diagnosis is made. Therefore clinical suspicion is the clue to the diagnosis, and performing the correct test will finally confirm the diagnosis.

- Another important differential diagnosis besides LP and LS in a case presenting with oral and genital erosions like the case presented would be some type of pemphigus (pemphigus vulgaris, paraneoplastic pemphigus).

- Although the histologic finding of subepidermal blisters or clefts is suggestive of MMP it can be seen occasionally in patients with lichen planus, lichen sclerosus, or even pemphigus, in particular in advanced lesions.

- Histology is considered less sensitive and specific for the diagnosis of MMP compared to DIF. It is more useful to rule out other diseases, but in many cases, it shows only non‐specific findings that cannot differentiate it from other diseases.

- The authors should review the newest guidelines for the diagnosis and management of MMP (Schmidt E, et al . European Guidelines (S3) on diagnosis and management of mucous membrane pemphigoid, initiated by the European Academy of Dermatology and Venereology - Part II. J Eur Acad Dermatol Venereol. 2021 Oct;35(10):1926-1948.). According to these new guidelines, the diagnosis of MMP is based on clinical findings together with the detection of anti‐basement membrane zone autoantibodies on the tissue by DIF (or direct immunoelectron microscopy), or circulating by indirect IF, ELISA or immunoblotting. Histopathology may be used in some cases as a complementary diagnostic tool. 

Author Response

Response to Reviewers’ Comments

Thank you for giving us the opportunity to submit a revised draft of the manuscript “An Underrecognized Histologic Clue to the Diagnosis of Cicatricial Pemphigoid: A Case Report and Review of Diagnostic Guidelines” for publication. We are grateful for the reviews provided by each external editor and appreciative of the time that each reviewer dedicated to providing insightful feedback on our manuscript.  The comments are encouraging and the reviewers appear to share our judgment that this case and its findings are clinically important. We have incorporated the suggestions made by the reviewers. Please see below, in red, our detailed response to each comment. All page numbers refer to the manuscript file with tracked changes. 

Responses to Reviewer 2:

In this paper, the authors report the case of a woman with mucous membrane pemphigoid (MMP) that was initially difficult to diagnose and propose some histologic clues for diagnosis, although the diagnosis was finally confirmed by DIF.

I have the following comments about this manuscript:

 - Mucous membrane pemphigoid (MMP) is NOT a type or variant of cicatricial pemphigoid (CP). In fact, they are synonymous, but some years ago it was decided to use the term MMP instead of CP, as not all cases/locations present with scarring.

Response: Thank you very much for this clarification. We have changed “MMP is a variant of CP” to “MMP, also called CP…” in all places where both MMP and CP are mentioned (page 1, line 14 and page page 1, line 33).

- Patients with MMP can target other antigens besides BP180 (type XVII collagen), and BP230 antigens (the terms BPAG1 and BPAG2 are not currently in use) including laminin 332, collagen VII, or integrin alfa6beta4.

Response: Thank you. We have added 5 additional BMZ antigens that can be targeted in other MMP subtypes: laminin-6, type VII collagen, integrin α6/β4 (oral and ocular MMP), p200 (anti-laminin gamma-1 pemphigoid), and laminin 332 (anti-laminin 332 MMP) (page 2, lines 49-52).

We also replaced “BPAG1” with “BP230” and “BPAG2” with “BP180; type XVII collagen,” both in the text (page 2, line 47) and in Figure 1.

- Many patients with MMP are misdiagnosed or not diagnosed initially due to the lack of knowledge about the disease because it is rare. It is not unusual to see patients that have been seen by many physicians (general practitioners, dentists, ENT, gynecologists, etc.) before a diagnosis is made. Therefore clinical suspicion is the clue to the diagnosis, and performing the correct test will finally confirm the diagnosis.

Response: We completely agree with this comment. We do state that diagnosis of MMP “should be based on clinical presentation and histology” (page 3, lines 84-85). However, to further emphasize this point, we added an additional sentence to the discussion (page 4, lines 138-41: “In such cases, we again emphasize the importance of clinical suspicion…correct diagnosis”).

- Another important differential diagnosis besides LP and LS in a case presenting with oral and genital erosions like the case presented would be some type of pemphigus (pemphigus vulgaris, paraneoplastic pemphigus).

Response: Thank you—we agree certainly that that is an important clinical differential. In this case, the histology revealed no evidence of pemphigus, so, in the interest of brevity, the manuscript did not discuss it in detail.

- Although the histologic finding of subepidermal blisters or clefts is suggestive of MMP it can be seen occasionally in patients with lichen planus, lichen sclerosus, or even pemphigus, in particular in advanced lesions.

Response: Thank you—this is a very important clarification to make. In our original submission, we used the term “subepidermal clefts,” when a more accurate phrasing is “subepithelial clefts along adnexae.” That change has been made. As you point out, the former can be found in a broad range of diseases. But, the latter can be fairly specific for a diagnosis of MMP in this setting, as Dr. Bernard Ackerman first described some decades ago, and as discussed further in the citation. We believe this clue is nevertheless  underrecognized. And, we have added this clarification to page 4, lines 143-46 (“Lastly…in this setting).

- Histology is considered less sensitive and specific for the diagnosis of MMP compared to DIF. It is more useful to rule out other diseases, but in many cases, it shows only non‐specific findings that cannot differentiate it from other diseases.

 Response: Thank you—we have added this information to the discussion (page 4, lines 131-32).

- The authors should review the newest guidelines for the diagnosis and management of MMP (Schmidt E, et al . European Guidelines (S3) on diagnosis and management of mucous membrane pemphigoid, initiated by the European Academy of Dermatology and Venereology - Part II. J Eur Acad Dermatol Venereol. 2021 Oct;35(10):1926-1948.). According to these new guidelines, the diagnosis of MMP is based on clinical findings together with the detection of anti‐basement membrane zone autoantibodies on the tissue by DIF (or direct immunoelectron microscopy), or circulating by indirect IF, ELISA or immunoblotting. Histopathology may be used in some cases as a complementary diagnostic tool.

Response: We agree and have added the following clarifying sentence in the discussion: “In this case, the MMP diagnosis was secured by both clinical findings and DIF studies, and was confirmed retrospectively by histopathology. In future cases, especially those without DIF studies, subepithelial clefting associated with adnexae could be a useful clue in this context.” (Page 4, lines 149-53).

Reviewer 3 Report

Thank you for your work on this important topic, particularly highlighting MMP vulvar/perianal disease as genital dermatoses are an understudied area in dermatology and dermatopathology. I have a few suggestions that, if incorporated, could improve the paper 

Line 93 - Is this the referenced granulation tissue featured in Figure 4? If so, I am not seeing the granulation tissue. Perhaps rewording is necessary or a histologic photo to highlight this granulation tissue would be helpful.

You also mention LS and d-VIN were ruled out. Was LP also ruled out at this time? What was the initial clinical differential diagnosis at the time of biopsy? Based on the clinical I would also include HSV in the ddx. Were the lesions swabbed for HSV PCR? 

Line 71 - For consensus on where to biopsy seems to be no difference in single site mucosal versus mucosal and skin (is sensitivity higher for mucosal or skin, or simply the more inflamed site?)

Line 104 - Remove the template words 

Line 123 - Referencing Figure 4 you state the biopsy shows predominantly granulation tissue. I do not see this from the histology provided (seeing erosion with pauci-cellular mixed inflammation and focal sub epidermal cleating). Would clarify this. 

Figure 4 - I cannot see where on the low power photo, the higher power images were taken (the high power photos seem to be more non-mucosal hair-baring sites vs the low power photo seems predominantly mucosal)

Thank you again for your work on this topic! 

Author Response

Response to Reviewers’ Comments

Thank you for giving us the opportunity to submit a revised draft of the manuscript “An Underrecognized Histologic Clue to the Diagnosis of Cicatricial Pemphigoid: A Case Report and Review of Diagnostic Guidelines” for publication. We are grateful for the reviews provided by each external editor and appreciative of the time that each reviewer dedicated to providing insightful feedback on our manuscript.  The comments are encouraging and the reviewers appear to share our judgment that this case and its findings are clinically important. We have incorporated the suggestions made by the reviewers. Please see below, in red, our detailed response to each comment. All page numbers refer to the manuscript file with tracked changes. 

Responses to Reviewer 3 Comments:

Thank you for your work on this important topic, particularly highlighting MMP vulvar/perianal disease as genital dermatoses are an understudied area in dermatology and dermatopathology. I have a few suggestions that, if incorporated, could improve the paper

Line 93 - Is this the referenced granulation tissue featured in Figure 4? If so, I am not seeing the granulation tissue. Perhaps rewording is necessary or a histologic photo to highlight this granulation tissue would be helpful.

Response: Thank you. We have altered this to “late-phase granulation tissue and fibrosis,” (lines 92-93), a more accurate description. Also, we discovered that Figure 4 (histopathology) and Figure 3 (DIF) were switched; that labeling has been corrected, as have the call-outs in the text.

You also mention LS and d-VIN were ruled out. Was LP also ruled out at this time? What was the initial clinical differential diagnosis at the time of biopsy? Based on the clinical I would also include HSV in the ddx. Were the lesions swabbed for HSV PCR?

Response: Thank you for pointing this out. The initial DDx at the time of the first (lesional) biopsy was MMP, LP, LS, and d-VIN (we added this to page 2, lines 93-95). LP was indeed ruled out on biopsy (we added this to line 95). We agree that HSV would be in the clinical differential. No evidence of HSV was found on biopsy, and the lesions did respond to steroids. Thus, since the clinical, DIF, and histopathology supported MMP, and in the interest of brevity, the manuscript does not discuss the differential of HSV in detail. The lesions were not swabbed for HSV PCR.

Line 71 - For consensus on where to biopsy seems to be no difference in single site mucosal versus mucosal and skin (is sensitivity higher for mucosal or skin, or simply the more inflamed site?)

Response: Thank you for pointing this out—the cited paper (PMID: 11902988) does not state whether sensitivity is higher for mucosal or for skin. You are correct; there was no stated difference in the ideal biopsy site and technique for single site mucosal vs. mucosal and skin.

Line 104 - Remove the template words

Response: Thank you; the template words have been removed.

Line 123 - Referencing Figure 4 you state the biopsy shows predominantly granulation tissue. I do not see this from the histology provided (seeing erosion with pauci-cellular mixed inflammation and focal sub epidermal cleating). Would clarify this.

Response: Thank you. In the figure legend, we altered this to “fibrosis and late-phase granulation tissue,” a more accurate description. Also, we discovered that Figure 4 (histopathology) and Figure 3 (DIF) were switched; that labeling has been corrected, as have the call-outs in the text.

Figure 4 (now figure 3)- I cannot see where on the low power photo, the higher power images were taken (the high power photos seem to be more non-mucosal hair-baring sites vs the low power photo seems predominantly mucosal)

Response: Thank you for pointing this out—images A, B, and C are from different areas within the same biopsy.

Thank you again for your work on this topic!

Round 2

Reviewer 2 Report

The manuscrit has been improved compared to the original one. However I have the following comments:

- “The anti-BMZ autoantibodies produced by MMP patients typically bind to BP230’s C-terminal domain, whereas those produced by BP patients typically bind to BP230’s NC16A domain”- There is a mistake- It should read BP180.

The authors use the terms CP and MMP indifferently in the text. They should use only one, and MMP should be preferred as it is the denomination that is currently used.  

- The authors should cite the recent Diagnostic Guidelines of MMP: Schmidt E, Rashid H, Marzano AV, Lamberts A, Di Zenzo G, Diercks GFH, Alberti-Violetti S, Barry RJ, Borradori L, Caproni M, Carey B, Carrozzo M, Cianchini G, Corrà A, Dikkers FG, Feliciani C, Geerling G, Genovese G, Hertl M, Joly P, Meijer JM, Mercadante V, Murrell DF, Ormond M, Pas HH, Patsatsi A, Rauz S, van Rhijn BD, Roth M, Setterfield J, Zillikens D, C Prost, Zambruno G, Horváth B, Caux F. European Guidelines (S3) on diagnosis and management of mucous membrane pemphigoid, initiated by the European Academy of Dermatology and Venereology - Part II. J Eur Acad Dermatol Venereol. 2021 Oct;35(10):1926-1948. 

- The gold standard for the diagnosis of MMP is DIF. Although histologic cues on H&E may be useful, the paper should insist on that. At the present time, we cannot make a diagnosis and use aggressive therapies (prednisone, immunosuppressants,  rituximab) with an H&E diagnosis. Even if DIF is negative the correct approach woul be to repeat DIF. I recommend also citing the work of Shimanovich I, Nitz JM, Zillikens D. Multiple and repeated sampling increases the sensitivity of direct immunofluorescence testing for the diagnosis of mucous membrane pemphigoid. J Am Acad Dermatol. 2017 Oct;77(4):700-705.e3. 

Author Response

Response to Reviewers’ Comments (Second Round Edits)

Thank you very much for allowing us to resubmit our revised draft of the manuscript, “An Underrecognized Histologic Clue to the Diagnosis of Mucous Membrane Pemphigoid: A Case Report and Review of Diagnostic Guidelines.” We are extremely grateful for your time and feedback on this paper, and have incorporated all of your suggestions. Please see below, in red, our detailed response to each comment. Thank you very much again.

Responses to Reviewer 2 (Second Round Edits):

- “The anti-BMZ autoantibodies produced by MMP patients typically bind to BP230’s C-terminal domain, whereas those produced by BP patients typically bind to BP230’s NC16A domain”- There is a mistake- It should read BP180.

Response: Thank you very much—this change has been made.

- The authors use the terms CP and MMP indifferently in the text. They should use only one, and MMP should be preferred as it is the denomination that is currently used.  

Response: Thank you very much—we agree and have made this change throughout the manuscript, including in the title.

- The authors should cite the recent Diagnostic Guidelines of MMP: Schmidt E, Rashid H, Marzano AV, Lamberts A, Di Zenzo G, Diercks GFH, Alberti-Violetti S, Barry RJ, Borradori L, Caproni M, Carey B, Carrozzo M, Cianchini G, Corrà A, Dikkers FG, Feliciani C, Geerling G, Genovese G, Hertl M, Joly P, Meijer JM, Mercadante V, Murrell DF, Ormond M, Pas HH, Patsatsi A, Rauz S, van Rhijn BD, Roth M, Setterfield J, Zillikens D, C Prost, Zambruno G, Horváth B, Caux F. European Guidelines (S3) on diagnosis and management of mucous membrane pemphigoid, initiated by the European Academy of Dermatology and Venereology - Part II. J Eur Acad Dermatol Venereol. 2021 Oct;35(10):1926-1948. 

Response: Thank you for providing such a thorough and comprehensive article on MMP diagnostic guidelines. We have added an additional paragraph to the discussion, which cites this article several times (particularly in reference to the comment below).

- The gold standard for the diagnosis of MMP is DIF. Although histologic cues on H&E may be useful, the paper should insist on that. At the present time, we cannot make a diagnosis and use aggressive therapies (prednisone, immunosuppressants,  rituximab) with an H&E diagnosis. Even if DIF is negative the correct approach woul be to repeat DIF. I recommend also citing the work of Shimanovich I, Nitz JM, Zillikens D. Multiple and repeated sampling increases the sensitivity of direct immunofluorescence testing for the diagnosis of mucous membrane pemphigoid. J Am Acad Dermatol. 2017 Oct;77(4):700-705.e3. 

Response: We completely agree—we have added a paragraph to the discussion that emphasizes this point. Thank you for the additional article regarding the sensitivity of multiple, repeated biopsies for DIF. We also added a sentence to the conclusion that further reinforces this point: “Importantly, this histopathologic picture remains secondary to DIF testing for anti-BMZ autoantibodies, the gold standard for the diagnosis of MMP” (lines 185-187).

Reviewer 3 Report

I think reviewer comments have been addressed and the manuscript is now suitable for publication. Thank you.
